# Flexible Framework to Model Industry 4.0 Processes for Virtual Simulators

**Kiara Ottogalli** [1,2,*], **Daniel Rosquete** [1,2,*], **Aiert Amundarain** [1,2,*], **Iker Aguinaga** [1,2,*] and **Diego Borro** [1,2,*]

1  Ceit, Manuel Lardizabal 15, 20018 Donostia/San Sebastián, Spain
2  Universidad de Navarra, Tecnun, Manuel Lardizabal 13, 20018 Donostia/San Sebastián, Spain
*  Correspondence: kottogalli@ceit.es (K.O.); drosquete@ceit.es (D.R.); aamundarain@ceit.es (A.A.); iaguinaga@ceit.es (I.A.); dborro@ceit.es (D.B.)



**Featured Application: Aircraft final assembly line assisted by collaborative robotics, a high-voltage cell simulator, and a machine tool simulator.**

**Abstract:** Virtual reality (VR)- and augmented reality (AR)-based simulations are key technologies in Industry 4.0 which allow for testing and studying of new processes before their deployment. A simulator of industrial processes needs a flexible way in which to model the activities performed by the worker and other elements involved, such as robots and machinery. This work proposes a framework to model industrial processes for VR and AR simulators. The desk method was used to review previous research and extract the most important features of current approaches. Novel features include interaction among human workers and a variety of automation systems, such as collaborative robots, a broader set of tasks (including assembly and disassembly of components), flexibility of modeling industrial processes for different domains and purposes, a clear separation of process definition and simulator, and independence of specific programming languages or technologies. Three industrial scenarios modeled with this framework are presented: an aircraft assembly scenario, a guidance tool for high-voltage cell security, and an application for the training of machine-tool usage.

**Keywords:** framework; process simulation; virtual manufacturing; virtual reality; augmented reality

## 1. Introduction

The integration of new information and automation-based technologies in industry is leading to a revolution, known as Industry 4.0, which is profoundly changing industrial processes and the workplace itself. The concept of Industry 4.0 was introduced in Germany in 2011 with the presentation of an initiative named "Industrie 4.0" [1]. Although there is no consensus on its definition, an important feature of Industry 4.0 is the integration of new technologies (such as the Internet of Things (IoT), cyber-physical systems (CPS), and virtual reality (VR)) into industry, enabling flexibility, robustness, and adaptability over time to cope with a changing business environment [1–3]. It is considered the fourth industrial revolution and may deeply change the future of manufacturing and production processes, leading to smart factories and interconnected industrial environments [4].

In this context, production becomes highly configurable and industrial processes change rapidly in order to adapt to changes in demand. VR is a key technology adopted in Industry 4.0 to provide an efficient and cost-effective way to model industrial processes. Research and development carried out by both the academic community and industrial companies has made VR mature, stable, and usable. This technology has proved to be effective in many industrial applications such as design, virtual prototyping, decision making, virtual assembly, training, manufacturing, ergonomics and knowledge

capture, and process planning and simulation [5,6] in different fields of engineering such as automotive, aerospace, and mechanical engineering, among others [7].

Augmented reality (AR) is rapidly evolving too. AR is a technology that allows for the overlaying of digital information on the real world and hence augmenting a real environment with computer-generated information and objects that are displayed in real-time and may be dependent on the real environment itself (context-sensitive) [8]. An example of AR in an assembly operation is a text displaying the next task that a worker has to perform and an arrow pointing to the next part to assemble. It has been used recently in many different areas of knowledge such as medicine, construction, and industry. In the last case AR is being used to support process monitoring [9], human-robot interaction/cooperation [10,11], maintenance [12–14], and assembly [15–17].

Planning an industrial process is normally a very complex and time-consuming process. Usually it is an iterative process prone to finding problems that cannot be predicted until the entire process is implemented. Making a virtual simulation of the process first can help to identify different issues beforehand, preventing unexpected costs and time-loss in advance. This is especially true in the case of immersive process simulators that can take advantage of the feedback provided by workers or validation engineers.

In Industry 4.0, a simulator has to incorporate not only the human factor, as this is of great importance in planning, testing, and improving processes [6], but also all the automation systems that interact with humans, such as collaborative robots. Contrarily to most VR-based simulations that are fixed, an industrial simulator must be flexible enough to cope with the fast-evolving needs of industry. For these reasons, there is a need for a structured way to model the industrial processes within a simulator considering three relevant factors: flexibility, to adapt to different purposes and application domains, and also to allow faster development; interaction of different actors (human and automation systems); and appropriate feedback to assist users throughout the entire process.

The goal of this research is to present a new framework that focuses on modeling Industry 4.0 processes for virtual simulators. This framework encompasses former research features such as task modeling, modularity, reusability, real-time data monitoring and user feedback, and novel features such as interaction among a variety of actors present in Industry 4.0 scenarios (besides human workers) like robots, sensors, and other systems, a broader set of tasks (including assembly and disassembly), a clear separation between process definition and simulator, and the flexibility to be used in different domains and adapt to multiple technologies.

By considering the most relevant features from previous works and adding new ones, the resulting framework proposes a solid groundwork upon which to allow faster development of new simulators, adaptation to different simulation scenarios, and, hence, the ability to cope with fast-changing industrial processes and satisfy the need for cost-effective solutions for testing a process before its actual implementation. Furthermore, it does not focus on the specifics of developing a simulator as model acquisition, model preparation (simplifying, texturing, or animating), or building an immersive environment (importing models, positioning objects, configuring interactions, or adding inverse kinematics).

In order to gain deeper insight into modeling Industry 4.0 processes for VR and AR simulations desk research was performed. The primary source used was Google Scholar, with the keywords used being "industrial process", "simulation", "modeling", "task modeling", "framework", "virtual reality", and "augmented reality". The results were analyzed and the most relevant features were extracted. To validate the framework, it was applied to the development of three different industrial simulations; this is discussed in more detail in Section 4.

The rest of the paper is structured as follows. The state of the art is presented in Section 2. The framework is described in Section 3. Section 4 shows three different industrial applications for which the framework has been applied. A discussion is presented in Section 5.

## 2. State of the Art

The modeling of tasks in simulators has been approached from many different perspectives in the literature. Some works, such as [18–21], describe VR systems for the simulation of product assembly. They focus on the methods and algorithms related to modeling the physics to assemble components of a product but not to modeling the assembly process itself (i.e., the tasks and subtasks composing the process). Nowadays, there are many different engines, such as Unity 3D, that already incorporate most of these capabilities, so the concern shifts from modeling the physics of the system to modeling the industrial processes themselves.

A framework called VR_MATE is presented in [22]. This framework is used for the analysis of maintainability and assembly, including the definition of the structure of the process. Boccaccio et al. [23] present a framework for displaying technical information on AR to replace printed piping and instrumentation diagrams. They use Unity3D and Vuforia to place clickable hotspots over an augmented diagram. Lee [24] presents a framework for large-scale manufacturing layout modeling and a material flow simulation based on the stochastic Petri net model to check reachability in the manufacturing chain and queuing theory for the simulation. This framework allows the creation of AR labels for manufacturing devices and their relationship within the manufacturing chain in real-time, but is specific for manufacturing facilities. A system to automatically generate a sequence of tasks to disassemble a product using its geometry is presented in [25,26]. Similar systems are used in [14,15] to guide a worker using a generated task sequence using AR. A framework focused on maintenance, assembly, and disassembly operations with automatic sequence planning is presented in [27]. Even if task sequence planning taking into account the geometric restrictions of a product is useful to simplify the definition of a process, it cannot determine other operations necessary to complete an industrial process that are not related to the geometry itself. Our framework can take advantage of an automated task sequence generator such as the ones mentioned before to simplify some steps of the process definition, but it also has the capabilities to model other types of tasks.

For example, an important factor to model in an Industry 4.0 process is the simulation of collaborative robots and their interactions with workers. A human–robot collaboration (HRC) VR scenario with simple task modeling is presented in [28], in which a backing film removal and prepreg laying tasks are simulated. The tasks are very specific to the application scenario and, therefore, there is still a need to model more general tasks including more diverse automated systems. Mourtzi et al. [13] present a system for providing maintenance instructions in a robotic industrial use case. In this system, reusable maintenance sequences are generated once to be reused when needed.

AR has also been used to simulate industrial systems with automated systems and human workers. Makris et al. [16] present an AR system which aids the user in an HRC assembly environment. An AR system able to communicate the intent of motion of a robot arm is described in [29]. The objective of the system was to study if the user was able to differentiate colliding and non-colliding motions though robot motion planning overlaid onto the real world. Maly et al. [30] present a study where the application focuses on assessing the best way to move a robot using AR. Here, the framework separates all the features related to the AR environment and the interaction, such as calibration, tracking and recognition of hand gestures, and visual aids, from the process definition. With this approach, any improvement in these areas can be applied to the AR application without affecting any previously defined processes. Similarly, a two-phase framework for AR assembly manual applications using AR is proposed in [31]. Even though this framework is focused only on AR assembly planning, it is also applicable to VR environments.

The following works take a step further in modeling complex processes by defining more elaborate data structures and relationships. Hierarchical structures for modeling assembly tasks are presented in [32,33], but these structures only consider assembly/disassembly tasks and do not take into consideration the modeling of other general tasks such as the movement of an automated guided vehicle (AGV).

Puig et al. [34] have developed a language to model general activities in a VR simulator. The language was intended to model common household activities in a 3D simulator. Other works such as [35,36] present other more general languages to describe complex scenarios. A simplified version of the languages to describe actions presented in these works is considered to be background for action modeling.

After reviewing research related to modeling Industry 4.0 processes for VR and AR simulations, the most relevant features were extracted:

- Flexibility: the possibility of adapting the framework/model/tool to different domains, purposes, or configurations.
- Presence of framework: if the research presented a framework.
- Task modeling: if the work presented task modeling. If present, two levels of task modeling were identified: specific if the task modeling was specific to a certain domain or purpose (for instance only assembly/disassembly tasks) and general if the task modeling could be applied to a variety of tasks other than assembly/disassembly.
- HRC: if the work considered human-robot collaboration.
- Multi-actor: if the work considered various (not-human) actors in the simulations.
- Modularity: if the work presented logically separated components.
- Monitoring: if the work is or could be used for monitoring processes.
- Multi-domain: if the research could be adapted to model processes for different domains, such as aeronautics, the automotive domain, and logistics, among others.
- Multi-purpose: if the work could be used for different purposes, such as design, prototyping, training, and monitoring, among others.
- Real-time: if the work considered real-time simulations.
- Reusability: if the work could be used as a base from which to develop other simulations.
- Simulation engine/manager independence: if the framework/tool depends on the usage of a certain simulation engine or manager. This feature is important as it allows task modeling and simulations to be independent of certain technologies, and thus a defined process can be reused for different simulators.
- User feedback: if the work considered giving user feedback.
- Technology: if the work was developed for a certain technology. The technologies presented in the works were categorized as none, if there was no relation to any simulation technology; non-immersive, if the work was related to simulation on a non-immersive device such as a PC or tablet; VR, if the work was related to VR simulation; and AR if the work was related to AR simulation.

Based on these features, a comparison among the different works is presented in Table 1.

As can be observed, none of the works presented considered all features at the same time or could be adapted to VR and AR simulations.

Another important factor found while analyzing research in this area was that most of the works presented simulations or tools specific to a certain domain and only a few presented general frameworks. Most of the simulations are not flexible enough to be used as a base for adapting to a new process in a different domain.

Most works are modular, run in real-time, and present user feedback. They can also be applied to multiple industrial domains for different purposes while providing user feedback. On the other hand, 33% of the works do not consider task modeling and 50% of them model specific tasks, mostly focusing on assembly/disassembly tasks. Furthermore, only a few consider HRC, but only one considers tasks where other actors besides humans and robotic arms interact (such as AGVs or other automation systems).

**Table 1.** This table presents a summary of the features considered and their presence in the works reviewed.

| | Fl | Fr | TM | HRC | MA | Mod | Mon | MD | MP | RT | Re | SI | UF | Tech |
|---|---|---|---|---|---|---|---|---|---|---|---|---|---|---|
| | | | | | | **Features** | | | | | | | | |
| [13] | | x | S | x | | x | x | x | | x | x | x | x | AR |
| [14] | | x | S | | | | | x | | | x | x | x | AR |
| [15] | | | S | | | x | | x | | x | x | x | x | AR |
| [16] | | | G | x | | | | x | | x | | | x | AR |
| [17] | | | N | | | x | | x | x | x | x | | x | AR |
| [18] | | | N | | | | | x | | x | | | x | VR |
| [19] | x | | N | | | x | | x | x | x | | | x | VR |
| [20] | x | | N | | | x | | x | x | x | | | x | VR |
| [21] | x | | N | | | x | | x | x | x | | | x | VR |
| [22] | | x | S | | | x | | x | x | x | x | | x | VR |
| [23] | | x | N | | | | | | x | x | x | | x | AR |
| [24] | x | x | S | | x | x | x | | | x | x | | x | AR |
| [25] | | | S | | | | x | | | | x | x | | NI |
| [26] | | | S | | | | x | | | | x | x | | NI |
| [27] | x | x | S | | | x | x | x | x | x | x | x | x | NI |
| [28] | | | S | x | | | x | | | x | | | x | VR |
| [29] | | | N | x | | | | | | x | | | x | AR |
| [30] | | | N | x | | x | | x | x | x | | | x | AR |
| [31] | x | x | S | | | x | | x | | x | x | x | x | AR |
| [32] | x | | S | | | x | | x | x | x | x | | x | VR |
| [33] | x | | S | | | x | | x | x | x | x | | x | VR |
| [34] | x | | G | | | x | | x | | x | | x | x | NI |
| [35] | | | G | | | | | x | x | | x | x | | N |
| [36] | | | G | | | | | x | x | | x | x | | N |

Legend: Fl, flexible; Fr, framework; TM, task modeling (N, not present; S, specific; and G, general); HRC, human–robot collaboration; Mod, modularity; Mon, monitoring; MA, multi-actor; MD, multi-domain; MP, multi-purpose; RT, real-time; Re, reusable; SI, simulation engine/manager independence; UF, user feedback; and Tech, technology (N, none; NI, non-immersive; VR, virtual reality; and AR, augmented reality).

The framework presented in this paper focuses not only on assembly planning but also on the broader area of process planning. We separate the process definition from the simulator. This makes the framework modular and reusable and, thus, flexible enough for testing different process configurations on the same simulator or even for trying the same process on a simulator developed with other technologies. The framework also considers HRC with various levels of automation.

These features are important in the simulation of many industrial processes. Nowadays, these processes require the simultaneous interplay of workers, machinery, and other automation devices such as collaborative robots or AGV. Thus, the definition of a task requires detailed scheduling to make it safe and optimal. Our work proposes an organized solution and definition for this schedule. Also, using a process definition separated from the simulator allows every industry to develop its own solution with reusable modeling which is easily portable to any language, allowing an agile implementation of any process.

The framework has some limitations. Firstly, as it was conceived as a fast and simple way for modeling processes for industrial VR/AR simulators, it is not intended as a mathematical theory or formal language with which to model an industrial process. Secondly, the framework adapts to the actual level of expertise of the modeler, and thus the resulting process can vary in terms of organization and granularity. Thirdly, networking issues among the devices that may be used in an industrial AR simulation are not considered.

## 3. Framework Description

We propose a framework comprised of two modules and an interface between them (see Figure 1). The first module handles the process definition, in which the entire industrial process is defined and stored in a database. This includes the definition of the actors (i.e., one or more workers, robots, and machinery, etc.) that take part in the completion of the task. The actors considered can be real, such as several workers completing a task collaboratively or a control panel of a machine in an AR application, or virtual, such as a robot in a VR application. This makes it possible to have, for example, a simulator with a real worker controlling a virtual machine through a real device (as a control panel or haptics).

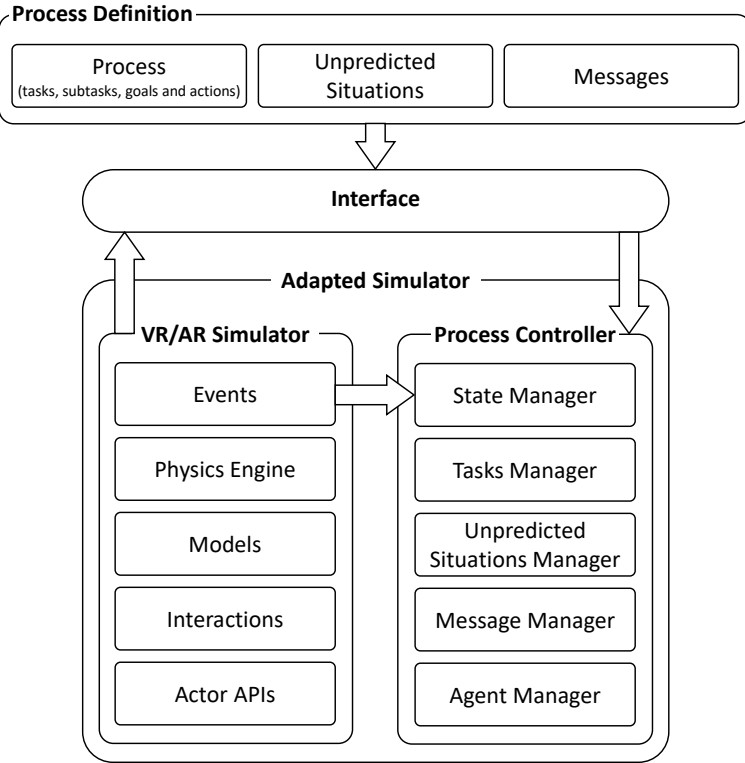

**Figure 1.** Framework scheme.

The second module is the simulator itself. Its core is the process controller that manages the state of the simulation, the flow of tasks that form the entire process (including the handling of unpredicted situations (i.e., an AGV running out of battery)), and the feedback to the user. Here, real or virtual automated actors (such as robots and AGVs, among others) define a series of predefined actions through a programmatic Application Programming Interface (API).

The interface connects the actions defined in the process definition with the actions defined by the APIs of the automated actors present in the simulator and passes them on as inputs to the process controller.

### 3.1. Process Definition

We define an industrial process as a group of interrelated high-level tasks, with each one comprised of interrelated subtasks. This definition is intended to be general for all kinds of processes, including the ones for processes and discrete manufacturing.

For instance, consider a scenario where a worker is assembling a certain product and a robot is used to bring a new box of screws from storage five meters away once the previous box is empty. In this scenario, a task would be a high-level operation such as "replace an empty box of screws".

This task is divided into several subtasks: "grab the empty box", "go to the storage area", "find a place to place the empty box", "leave the empty box", "find a box with screws", "grab the box of screws", "go to the working area", and "leave the new box of screws". The granularity of the tasks depends on the needs of the simulation to be modeled.

The structure of a task contains an information field, dependencies, and subtasks. The information field defines its general purpose, such as, for example, "replace an empty box of screws". This serves as a general description of the set of subtasks that involves changing the empty box for a new one.

The dependencies are the set of tasks that need to be completed before the task begins. At any given time, a task can depend on the finalization of one or more tasks and it may be executed in parallel with other independent tasks. These dependencies may be modelled as a hierarchical structure of the tasks and subtasks that define a process. This structure is an extension of the assembly models presented [32,33], since our subtasks are not restricted to assembly actions of parts.

Each subtask consists of an information field, its dependencies, a mode, atomic actions, and goals. The information field allows for guiding the users while performing the subtasks and providing feedback when the task is being completed by an automated system (i.e., a robot, an AGV, or another system). This information is stored as a message that has a type (info or error) and a localized description of the error.

As in the case of the tasks, the dependencies of a subtask are other subtasks that need to be completed before it starts. These are the implicit pre-conditions of the subtask.

The mode defines the automation level of the subtask. We define two different levels of automation: *manual* and *auto*. In *manual* mode, the human worker manually performs a subtask. While in *auto* mode, the system automatically executes a subtask using one of the predefined actors (such as a robot or machine). This mode allows for an optional fast-forward feature that performs a virtual simulation to complete the automatic operations instantaneously (this mode is helpful when the user needs to reach a certain point in the simulation while avoiding waiting for the entire completion of the operations).

To define the action of a subtask, we used a simplified version of the language defined in [35,36]. Each action requires an actor, which may be human or an automated system, which is in charge of executing the action. The actor is identified using an ID, which will be later used by the interface to map it within the simulator. A set of parameters are used to model targets, tools, and operation configuration.

The goals of a subtask are the conditions that need to be satisfied to consider the subtask as completed. When a subtask is completed, it triggers an event to change its global condition. The global condition triggers another event to notify all the subtasks that depend on it of its new value. In this way, other subtasks can begin if their conditions are met. This process is the core of the sequential model of the proposed framework.

Based on the previous definitions, we define a data model comprised by six classes: process, task, subtask, action, goal, and message (see Figure 2).

This model defines:

- A process that is composed of one or more tasks.
- The tasks and subtasks (with their actions and goals) that comprise the industrial process. The handling of unpredicted situations that may arise during the process is modeled as a set of tasks and subtasks.
- The localized message data containing the descriptions of all the tasks and subtasks.

As can be seen in Figure 2, a task may depend on other tasks to start and serve as a logical container for the subtasks. For this reason the task will not have actions or explicit goals, the implicit goal of the task being that it finalize its subtasks. Additionally, the subtasks may depend on other subtasks to start. Another point of the model is to allow parallelism among tasks and subtasks, because tasks or subtasks that are not dependent on one another start in parallel.

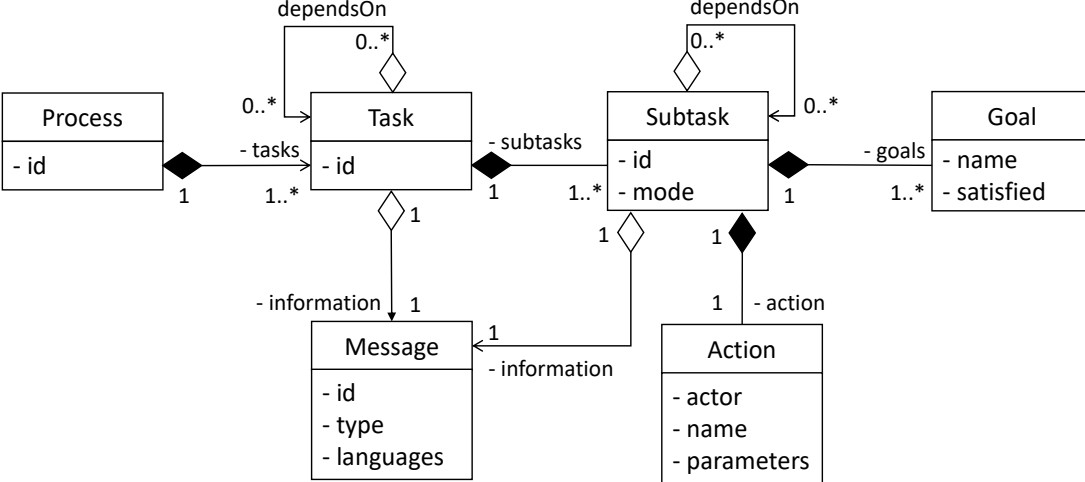

**Figure 2.** UML class diagram of the process data model.

Every subtask has an action in which the actor that will be responsible for its execution is defined. This, together with the consideration of the actor's operations by the interface (which will be explained in Section 3.3), allows the framework to deal with simulations which have multiple instances of different types of actors.

It is important to note that this is a base model and it can be extended during the design phase depending on the requirements of the simulation. The decision of using this model and no other models presented in the bibliography is based on its simplicity, and, therefore, the person in charge of modeling the process does not need to be an expert in a certain mathematical model/theory (e.g., petri nets) or a more formal modelling language (e.g., Business Process Execution Language (BPEL)), but may adapt the model to the theory, language or format of his or her expertise that he or she finds suitable to use for the industrial process to be defined. This decision is aimed at making the process definition more flexible and less time consuming.

The data can be stored in any database structure and is processed later by the interface to generate the input needed by the process controller adapted to the simulator.

### 3.2. Process Controller Adapted to the Simulator

Our framework, which can be used to model an industrial task, is flexible enough to support different technologies used for the simulator itself, such as VR or AR. In the case of a VR simulator, it runs a physics engine that contains the structure of the virtual scenario and its physical relationships. This includes which parts are fixed and which ones are moveable and the kinematical relationship between them. In the case of tasks that are automated (i.e., ones which will be completed by a robot) it also includes the robot kinematics. An AR simulator includes the tracking system, hand gesture recognition, and visual aids.

The process controller is responsible for keeping the state of the simulation, controlling the flow of tasks and subtasks of the process simulator, managing unpredicted situations, giving feedback to the user, and monitoring different aspects of the process. It manages four submodules: the state manager, the tasks manager, the unpredicted situations manager, and the message manager. In some simulators, it also communicates with a special monitoring module called the agent manager.

#### 3.2.1. State Manager

The state manager is a centralized way to store the state of the entire simulation. The state of the simulation is defined by all the variables that are relevant during the industrial process. These variables are named the global conditions.

A global condition is an independent requirement related to the achievement of a certain action. It is simply defined by a Boolean variable that expresses the state of an entity inside the simulation.

Each global condition is related to one or more conditions (goals), each one belonging to a subtask. Each goal listens to its related global condition, waiting for a state change. If the value of the global condition is the same as the value required by the goal of the subtask, the goal is considered achieved. A subtask is complete once all its goals are achieved.

Figure 3 presents a schematic example of the operation of the state manager. In this case, an action performed in the simulation environment triggers an event. A global condition, $a$, which is listening to this event switches its value (from false to true), changing the state of the simulation and triggering another event that checks that its associated condition $a$ on the current subtask, $Subtask_0$, has been satisfied. In this case, when both conditions $a$ and $c$ on $Subtask_0$ are satisfied, the subtask is considered complete and it finishes, allowing $Subtask_1$ to begin.

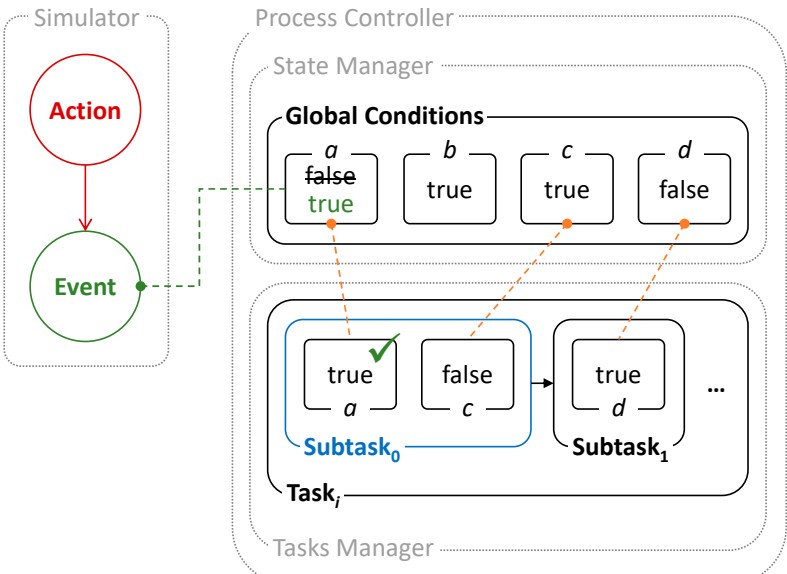

**Figure 3.** State manager operation.

The first step to adapt the process controller to a specific simulator is to define the set of global conditions and link them to the events of the entities inside the simulator. Thus, when an action changes the status of an entity, an event is triggered, changing the global condition related to it. These global conditions are defined in the process definition (see Section 3.1).

### 3.2.2. Task Manager

The task manager is in charge of coordinating all the tasks and subtasks inside the simulation, making automated actors perform different procedures and aiding human actors while performing actions.

Tasks and subtasks have five statuses: *waiting*, *ready*, *running*, *completed*, and *cannot_complete* (see Figure 4). When a subtask depends on the finalization of another subtask (parent subtask) and the latter has not finished, it is in a *waiting* state. When all the parents of a subtask have finished, then the state of the subtask changes to *ready*, which means that it may perform its action. As the goals of the subtask are satisfied, it enters a *running* state. Finally, when all the goals of the subtask are satisfied, it achieves its *completed* state. If there is an error related to the action performed, the subtask obtains the status of *cannot_complete*. If the error related to the action is solved, then the subtask shifts to a *completed* state.

The task manager listens to the events of the state manager through the goals inside the subtasks.

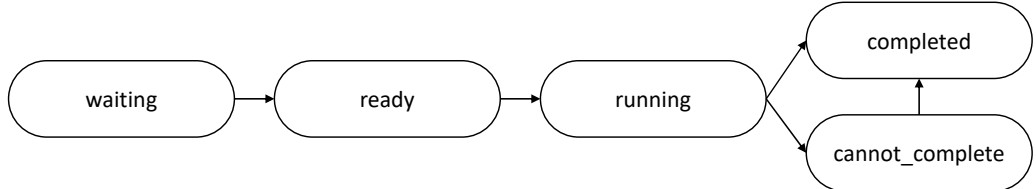

**Figure 4.** State machine of a subtask status.

Once an entity in the simulator (such as an actor, a collision detected by the physics engines, or when a user releases a trigger or performs a given motion . . . ) changes its status, it triggers an event (such as $E_1$ in Figure 5) which changes the related global condition in the state manager. This triggers the event $E_2$, where the related condition (goal) is checked for satisfaction. If it is satisfied, an event $E_3$ is triggered to make the current subtask verify its new status.

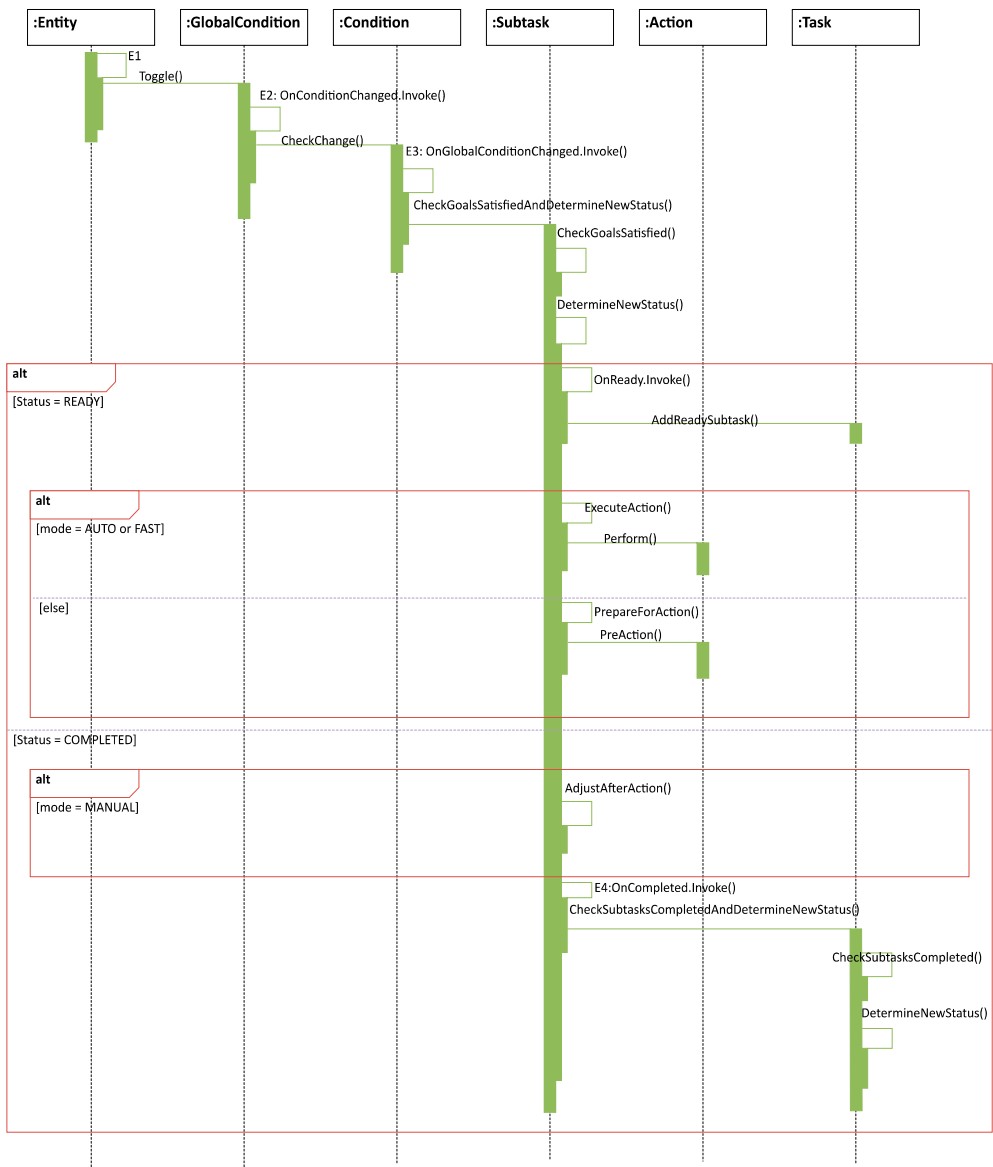

**Figure 5.** UML sequence diagram of the task manager.

When a subtask is complete, an event $E_4$ is triggered to notify its parent task that it has finished. If there are no more subtasks in a waiting state inside the current task, the task is considered complete.

If a subtask is in a *waiting* state depending on the finished subtask, it will change its state from *waiting* to *ready*. The task manager will repeat this management procedure until all tasks of the process are complete.

### 3.2.3. Unpredicted Situations Manager

The unpredicted situations manager is in charge of receiving notifications from the simulation about situations outside the normal flow of the process. Once it receives this notification, it is responsible for checking if the error has already been notified so that it is not handled twice, and if it updates another error (in the case of errors that are mutually exclusive), then it includes the error in a queue to be handled and triggers an event that will be listened to by the task manager, which will handle the error.

When the error is correctly handled by the task manager, the unpredicted situations manager will get a notification and remove the error from the queue.

Unpredicted situations can arise at any point during a process. We consider the handling of these situations as a branch in the workflow of the tasks. When an unpredicted situation arises, the unpredicted situations manager catches this error and adds it to a queue for handling. Then, it notifies the tasks manager, which will immediately divert the simulation flow of tasks to the process defined in the process definition for handling the unpredicted situation.

When the task manager finishes handling the unpredicted situation, it will restart the normal flow of the process and notify the unpredicted situations manager.

The Task Manager does not allow any actor to continue with the process if one or more of the conditions have not been satisfied, or if an error has occurred.

### 3.2.4. Message Manager

The message manager is responsible for assisting the human actor during the realization of the process. It loads descriptive localized messages related to the tasks and subtasks being performed from the process definition and listens to the tasks manager events for the status of the tasks.

### 3.3. Interface

One of the purposes of this framework is to detach the process definition from the specifics of the simulator, making it modular and reusable once it has been defined.

This separation allows for abstracting the modeling of the industrial process from the operations and capabilities actually present in the actors. Thus, it makes the process of developing a simulator less cumbersome to the designers as they do not have to remember the specific API of each actor of the simulation or rewrite the same process for a similar actor with a different API.

Therefore, we propose an interface that matches actions defined in the process definition with the ones on the APIs of the actors. In this way, if an actor or the entire simulator changes, the only part which has to change is the interface. In addition, as the process is not hardcoded in the simulator, the interface allows for easily testing multiple process definitions of the same scenario to find out which one works best.

A suitable and simple data structure for this purpose is a dictionary. The interface consists of a dictionary that links each of the actions present in the process definition database to its counterpart in the actor APIs. The syntax of the actions must be the same as in the process definition database and the interface.

Once the dictionary is defined, the interface is able to interpret and match the action parameters stored in the process definition to the ones required by the corresponding actor API. For example, an automated actor instance $I$ should move from its current position $A$ to a new position $B$. The action defined in the process definition will have actor $I$, action $M$, and the positions $A$ and $B$ as parameters. The interface will match this definition of the action $M$ with the corresponding operation on the API of the $I$ actor, sending the $A$ and $B$ parameters in the proper format.

## 4. Results

Currently the proposed framework has been implemented using the JSON format for the process definition and the Unity3D engine, using C#, for the simulator. To validate the generality of the proposed framework it has been applied to the development of three industrial scenarios in VR and AR: an aircraft final assembly line, a high-voltage cell simulator, and a machine tool simulator. These scenarios have different devices, actors, and purposes.

### 4.1. Aircraft Final Assembly Line Simulator

This simulator was developed considering an aircraft final assembly line scenario for the aeronautics domain. The purpose of this simulator is to evaluate different assembly strategies with HRC and perform an ergonomics assessment.

Two features of the proposed framework were very important while developing this simulator: detaching the process definition from the simulator and considering various actors.

The first feature allowed for the implementation of only one simulator adapted to the environment, one interface, and for the modeling of different definitions for the processes to be evaluated. A total of 24 processes, including 13 totally automated processes (where the actors are only the automated systems) and 11 HRC processes, were implemented. This made it possible to evaluate different processes with which to assemble the cabin and cargo of an aircraft in automatic and semi-automatic ways on one simulator. The second feature allowed for the definition of tasks where the actions were performed by different actors (workers and robots) in parallel.

The resulting simulator is a VR immersive simulator based on HTC Vive (headset and controllers). Noitom's Perception Neuron motion capture system [37] was used for the ergonomics assessment in the aircraft assembly simulator.

The environment in this simulator consists of a slice of an aircraft placed inside a warehouse. The simulator is focused on new assembly processes in the cabin and cargo areas such as the assembly of the hatracks (upper luggage containers) and the wall linings in the cabin and the sidewall panels and for the cargo area.

The simulator considers different types of actors:

- Human actors, which are used to check the operation and push the parts for the final adjustment.
- AGVs, which are used to transport racks containing the parts and robots.
- Collaborative robot arms (KUKA LBR iiwa 14), which are a robot arms used to lift and assemble the sidewall panels and linings.
- Custom empowering arms, which are mobile robotic devices used to aid the worker while lifting and placing the hatracks.
- As an example, one of the subtasks of the process is to place a sidewall panel in its final position, once it is grabbed by the robotic arm, as can be observed in Figure 6a.

In the process definition, the placing subtask is modelled as

```
"id": 0,
"information": "SUBTASK_ROBOT_PLACE_SIDEPANEL_08_CABIN",
"dependsOn": [],
"mode": "auto",
"action": {
    "actor": "Robot02",
    "name": "place",
    "parameters": ["SidewallPanel08", "−90.0", "0.0", "0.0", "120.0", "180.0", "50.0", "−85.0"]
},
"goals": [ {"name": "BracketDown_SidewallPanel_08_Active", "satisfied": true}]
```

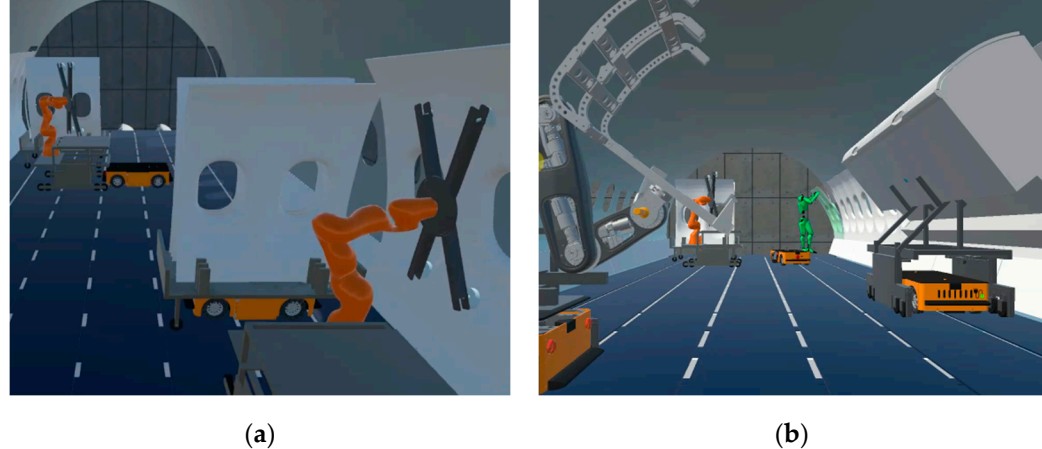

(**a**) (**b**)

**Figure 6.** Aircraft final assembly line simulation. (**a**) Robot arms placing sidewall panels while automated guided vehicles (AGVs) move the part racks. (**b**) HRC of a human and a robot arm placing sidewall panels, AGVs moving the part racks, and a custom robot moving to the place where it should lift the hatrack.

This defines the data of the subtask *id*, the information *id* to be managed by the message manager, the dependencies on other subtasks (in this case none), auto mode, the action, and the goals. The action indicates the name of the robotic arm (there could be many within the scenario), the natural language name of the subtask, and the final position of the specific sidewall panel. This subtask has a unique goal which will wait for a true state from the state manager. The interface will interpret and link the action to one operation defined on the API of the robot in the simulator and perform it. After this, when the robot arm places the sidewall panel in its final position, the task manager will be notified through the events triggered by the state manager. Finally, the task manager will initiate the next subtask.

Another option for the aircraft assembly process considers HRC where the robot arms, AGVs, and human are working on the cargo area at the same time (Figure 6b). Also, as can be observed, the worker shows a color representing one of the four risk categories of the Ovako Working posture Analysis System (OWAS) [38], which is used to perform real-time ergonomics assessment.

An important feature of many virtual reality simulators is their ability to monitor the run-time execution of the simulator through agents [39]. In this framework, an agent is any software module that can be connected to the simulator for monitoring purposes. This simulator has a module comprised of two agents that allow storing relevant data from the process in real time, but this may be extended as needed.

The first agent is in charge of monitoring the movements of selected entities inside the simulation. This agent is designed to store the pose (position and rotation according to a certain coordinate system) of the entity of interest in a database with the purpose of replaying the process later for further study, for instance to perform ergonomics assessment from motion data captured of a worker performing an assembly task.

The second agent captures and stores data concerning the initial and final time of the tasks and subtasks performed by the actors (human or automated) inside the simulation in order to find bottlenecks in the process as it is, the level of difficulty of different tasks, problems in synchronization, different courses of actions depending on the expertise of the workers, and comparing similar automated agents doing the same process.

## 4.2. High-Voltage Cell Simulator

A VR- and AR-based simulation was developed to train a worker on the actuation of a machine, in this case a high-voltage cell, where training with actual equipment is expensive and, in some cases, even dangerous. Some of the basic tasks used to train a worker are turning on, unlocking, changing

fuses, and starting the machining, among others. Using these kinds of applications results on reduced costs for the company, avoidance of misuse of machinery and reduced time of training.

The VR simulator has been adapted to multiple devices such as HTC Vive and Oculus Quest while parts of the simulator have been adapted to provide online guidance to workers using the Microsoft Hololens or Magic Leap devices. This shows one of the strengths of the approach proposed. By separating the definition of the tasks from the details of the simulators (such as the interaction details or the nature of the application) a high degree of flexibility is enabled.

To achieve the training, two actors are generally required:

- Human: able to perform many tasks with the machine panel.
- Machine: consists of a panel with many locks, keys, fuses, and an inside mechanism to actuate the machinery.

One of the tasks is to unlock a door containing the fuses (Figure 7). In this task, the worker has to put the switch lock in position and place a lug wrench in the proper place and turn it 45 degrees counterclockwise in order to open a fuse box. The turning subtask is modelled as

```
"id": 5,
"information": "SUBTASK_TURN_WRENCH_COUNTERCLOCKWISE",
"dependsOn": [4],
"mode": "manual",
"action": {
    "actor": "Worker01",
    "name": "turn",
    "parameters": ["LugWrench", "−45.0"]
}
"goals": [ {"name": "Wrench_On_Minus_45", "satisfied": true}]
```

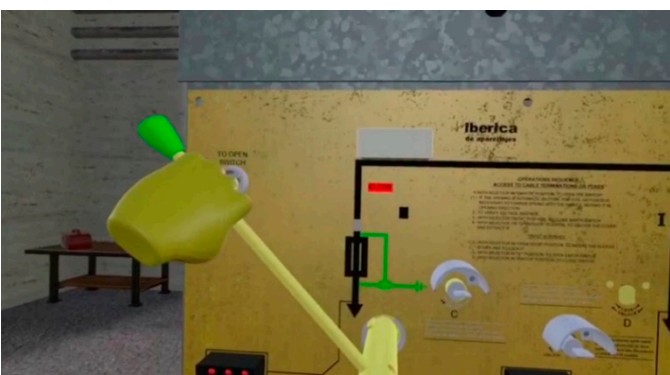

**Figure 7.** Unlocking a door containing fuses in the high-voltage cell simulation.

This subtask is similar to the one presented in Section 4.1. In this case the subtask has a dependency on the previous subtask, the mode is manual, the actor is worker 1 (there could be many within the scenario), the name of the action is "turn", and the parameters are the wrench which is used and the degree to which it is turned. Once the worker has manually performed this action, the fuse box will be unlocked in the simulation.

### 4.3. Machine Tool Simulator

The aim of this VR simulation is to train a worker on the procedure of a milling machine. Contrarily to the high-voltage cell simulator, in this case the machine has a complex user interface. The machine

itself is an automated Computer Numerical Control (CNC) lathe and milling machine with four degrees of freedom. Some of the basic tasks are turning on the machine, placing cutters, installing a gripper, fastening a part on the gripper, and milling this part in a precise place, among others. This application allows users to familiarize themselves with the machine and its procedures before accessing and using the real one, making it safer for the user and reducing machinery wear, energy consumption, and time of training.

The simulator is based on the HTC Vive immersive VR system while the embodiment of the user is based on Noitom's Perception Neuron motion capture system. The simulator, however, does not require this system to work.

To achieve the training, the simulator only requires two actors:

- Human: able to perform tasks with the machine panel.
- Machine: a machine with a control panel that allows for performing milling operations.
- One of the tasks present in this simulation is to install a milling cutter in the machine (Figure 8). To do this, the user has to grab the cutter, push a button to unlock the machine, and place the cutter into the correct position. The subtask to grab the milling cutter may be modelled as

```
"id": 0,
"information": "SUBTASK_GRAB_MILLING_CUTTER",
"dependsOn": [],
"mode": "manual",
"action": {
     "actor": "Worker01",
     "name": "grab",
     "parameters": ["MillingCutter"]
}
"goals": [ {"name": "Milling_Cutter_Grabbed", "satisfied": true}]
```

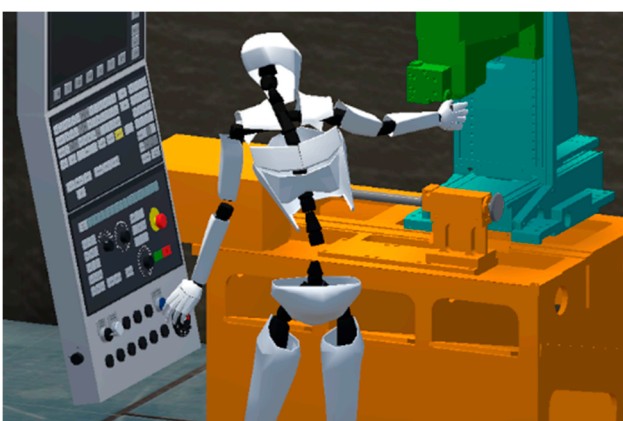

**Figure 8.** Installing a milling cutter in the machine tool simulation.

This subtask describes a manual action to be performed by worker 1 in which the goal is satisfied once the worker grabs the milling cutter.

## 5. Discussion

In this work a framework for modeling Industry 4.0 processes for VR and AR simulators has been presented.

This framework gathers the most important features present in previous works, such as modularity, reusability, consideration of HRC, task modeling, and independence of a certain simulation manager,

and new features, such as interaction among human workers and a variety of automation systems, a broader set of tasks (including assembly and disassembly), flexibility of modeling industrial processes for different domains and purposes, a clear separation of process definition and simulator, independence of specific programming languages, and VR/AR support.

The broad range of features present in the framework gives it the ability to easily include a wide range of features in the final simulator (ergonomic study, reproducibility, and worker assistance, among others).

As proof of concept, the framework was used to model three real industrial scenarios involving several elements typical of Industry 4.0, such as collaborative robots and automated machinery. The development of the simulators was completed on a specific platform (Unity3D). During the implementation of the simulators, we tested the features and capabilities of the proposed framework, including specific features, for each scene.

The framework proved to be flexible enough to allow modeling processes for different domains and purposes. For example, one of the simulators includes AGVs and robotic arms, while another is based on manual tasks following specific and ordered steps to turn on a machine. Regarding features, one of the simulators includes an ergonomics study, while another includes worker assistance displaying information on a panel. In the same way, one of the simulators (the aircraft final assembly line) included all the features mentioned in this article, obtaining a simulator and 24 different processes (13 totally automatic and 11 with HRC).

The ability to adapt to different platforms is also important. Two of the industrial scenarios were simulated in VR and adapted to the HTC Vive system; the third was also adapted to the Oculus Quest platform and some parts of it were adapted to provide online AR aid to workers with Microsoft Hololens or Magic Leap devices.

Another advantage found while using the framework for developing the three simulators presented was the rapid definition of multiple processes. The separation of process definition and simulator allows one to easily change similar actors in the simulation (e.g., a robotic arm) by simply modifying the interface without the need to change the process definition. Thus, testing multiple processes within a single simulator is allowed for. Using the framework, we were able to model three different industrial applications with different processes.

Although the framework simplifies the development of these simulators, it currently has some limitations. The framework is not intended as a formal/mathematical way to define a process but to present a fast and simple way to model industrial processes. It focuses on helping modelers make flexible and rapid simulations using their background and expertise. The framework is flexible about the modelling of the tasks, and for this reason, there is not a unique way of modelling a task. The way of dividing and organizing the tasks and subtasks inside a process is subjective and depends on the modeler.

There is a need to test different techniques within the process definition part of the framework. For the three industrial cases we presented in Section 4, we chose to define the processes in a JSON format and saving them as text files. These files serves as the input of the interface that makes the connection with the adapted simulator. It would be fruitful to identify, use, and compare other techniques for making the process definition.

Assuming this framework will be used in an AR environment which includes real devices such as robots, AGVs, and others working on the process, it is necessary to have a client-server network architecture where the server runs the adapted simulation and the clients are the devices. Our framework does not consider networking among devices, because for the scope of this study it was assumed to be fully operational, secure, and without delay, and, therefore, it is recommended that future research include this feature.

**Author Contributions:** Conceptualization, K.O. and D.R.; methodology, K.O. and D.R.; software, K.O. and D.R.; validation, K.O. and D.R.; investigation, K.O. and D.R.; resources, A.A., I.A., and D.B.; data curation, K.O. and D.R.; writing—original draft preparation, K.O., I.A., and D.R.; writing—review and editing, A.A., I.A., and D.B.;

visualization, K.O. and D.R.; supervision, A.A., I.A., and D.B.; project administration, A.A., I.A., and D.B.; funding acquisition, A.A., I.A., and D.B.

**Funding:** This research was developed within the following projects: SIMFAL (funded by H2020 Clean Sky 2, grant number 737881), Langileok (funded by the Basque Government, *Elkartek* program), and VARMAT (funded by Science, Education, and Universities Ministry, *RETOS-COLABORACIÓN* program).

**Conflicts of Interest:** The authors declare no conflict of interest.

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
