# Peer review of "Flexible Framework to Model Industry 4.0 Processes for Virtual Simulators"

_applsci, doi:10.3390/app9234983_

Round 1

Reviewer 1 Report

The topic of the paper is relevant and aligned with current trends (although sometimes over-emphasizing buzzwords) but the paper needs some improvements.

The English is reasonable (with a few mistakes) and the text in general reads well.

Abstract – should more explicitly emphasize what is new in this paper. Many of the used arguments are “traditional”, i.e. could be used with any simulation system from the past. Therefore, the authors need to say what is new.

Introduction:

“an important feature of industry 4.0 is the integration of new technologies into the industry” – be more precise: which new technologies? “AR allows the integration of digital information into the real world” – this sentence is not very clear / accurate. Be more precise. Paragraphs starting in line 70 and line 80 seem to have some overlapping. Better explain (make it explicit) what the research question is.

State of the art – some additional recent references should be considered. There are only few recent ones (apparently only 2 from 2018, none from 2019).

Make explicit the adopted research method.

“The first module handles the process definition ... using a specific language” – which language? BPMN/BPEL? If not, why not?

Fig. 1 is rather unclear – in Process definition you only mention “data” – no tasks or activities. Don’t you have activities in processes?

Also Fig. 1 does not show anything (explicitly) related to VR/AR ...

Section 3.1 – why not using a common formal language (e.g. BPMN/BPEL) to model the process?

Fig. 2: Only subtasks have Actions and Goals? What about Tasks?

Although the collaboration Human-Robot is mentioned several times, what is specific in the framework to cope with it?

Section 4.1 – it is IMPORTANT to highlight the VR features that make this simulator different from a classical simulator.  The authors fail to give convincing examples.

“The  aim  of  this  VR  simulation  is  to  train  a  student” – student or worker?

“the embodiment of the user is based on Noitom’s Perception Neuron motion capture system” – more details (or a reference) are needed.

“Although the framework simplifies the development of these simulators” – is the goal to develop “simulators” or to develop “simulation cases”?

In the discussion part it would be necessary to have some comparison with other (existing) systems, clearly highlighting the novel contributions.

It would also be important to mention the applicability advantages and how this could evolve to a real use in industry.

Author Response

Dear reviewer,

We would like to thank you for your effort in reviewing our paper. Your comments have served to improve the paper. Below you will find the revisions we have made according to your comments.

Abstract – should more explicitly emphasize what is new in this paper. Many of the used arguments are “traditional”, i.e. could be used with any simulation system from the past. Therefore, the authors need to say what is new.

Thanks to this comment we realized that the abstract could be more concise and moved some of it to the introduction and emphasized on what is new.

Introduction: “an important feature of industry 4.0 is the integration of new technologies into the industry” – be more precise: which new technologies?

We added a few examples of the new technologies integrated in industry 4.0.

“AR allows the integration of digital information into the real world” – this sentence is not very clear / accurate. Be more precise.

We changed the sentence to be clearer about what AR does.

Paragraphs starting in line 70 and line 80 seem to have some overlapping. Better explain (make it explicit) what the research question is.

This comment made us realize we were repeating some ideas in the two paragraphs. We fixed it and made the purpose of the research explicit.

State of the art – some additional recent references should be considered. There are only few recent ones (apparently only 2 from 2018, none from 2019).

We added additional references of works presented in 2019.

Make explicit the adopted research method.

We made the research method explicit in Section 1, and also pointed it out in the abstract thanks to reviewer 4 advice.

“The first module handles the process definition ... using a specific language” – which language? BPMN/BPEL? If not, why not?

You are right, this comment is confusing. The idea we were trying to explain is that the model is simple, adaptable to the background of the modeler and can be stored in any specific database format suitable to the specific project to be developed using the framework. We explain it better in section 3.1.

1 is rather unclear – in Process definition you only mention “data” – no tasks or activities. Don’t you have activities in processes? Also Fig. 1 does not show anything (explicitly) related to VR/AR ...

Based on your comment we realized that using “data” is too generic for what we are defining. We fix it. Also, we added the idea of VR/AR to the image.

Section 3.1 – why not using a common formal language (e.g. BPMN/BPEL) to model the process?

Answered in the seventh question.

2: Only subtasks have Actions and Goals? What about Tasks?

Yes, in our model only subtasks have actions and goals. The task is meant to be a logical container for subtasks. We explained it better in section 3.1.

Although the collaboration Human-Robot is mentioned several times, what is specific in the framework to cope with it?

The framework considers the multi-actor possibility for VR/AR simulation in the process definition, where defining the actor is important while defining the action, and in the process interface, where the APIs of multiple actors may be considered. We made it explicit in section 3.1.

Section 4.1 – it is IMPORTANT to highlight the VR features that make this simulator different from a classical simulator. The authors fail to give convincing examples.

The simulator presented in section 4.1 is similar to other simulators developed for the aeronautics domain, the difference is that it was developed using the proposed framework, and for this reason we were capable of defining 24 different scenarios defining different modules but using the very same simulator. We emphasized the features of the framework that made this possible, and gave it more importance in the first paragraphs of the section.

“The aim  of  this  VR  simulation  is  to  train  a  student” – student or worker?

We changed it to worker.

“Although the framework simplifies the development of these simulators” – is the goal to develop “simulators” or to develop “simulation cases”?

The goal of the framework is to model industry 4.0 processes for VR/AR simulators. We made it clearer on the abstract and Section 1. For this reason, we state that the framework made the development of these simulators easier.

In the discussion part it would be necessary to have some comparison with other (existing) systems, clearly highlighting the novel contributions.

We understand your concern, but as what we are proposing is a framework, to make a comparison, we made a compendium of features that we presented in the summary table and associated explanation in Section 2. We grouped the features already present and new features (as considering many automation systems, and tasks besides assembly/disassembly) in our framework. Furthermore, to validate the generality of our framework, we present 3 different industrial applications of our framework used as a base for modelling simulations in both, VR and AR (Section 4). We highlighted the novel contributions in Section 5.

It would also be important to mention the applicability advantages and how this could evolve to a real use in industry.

We added the applicability advantages in Section 5. Regarding real use in industry, the three simulators presented were developed for simulating real industry 4.0 scenarios.

Best regards,

Kiara Ottogalli, Daniel Rosquete, Aiert Amundarain, Iker Aguinaga and Diego Borro

Reviewer 2 Report

Paper is very good represented.

Authors has completed paper and is very good organized, starting from Abstract, Introductions, methods and representation of results. Conclusions are very clear represented.

Author Response

Dear reviewer,

We would like to thank you for your effort in reviewing our paper. Thank you very much for your comments.

Best regards,

Kiara Ottogalli, Daniel Rosquete, Aiert Amundarain, Iker Aguinaga and Diego Borro

Reviewer 3 Report

The revised manuscript answers to all of my queries and I recommend that the paper gets accepted

Author Response

Dear reviewer,

We would like to thank you for your effort in reviewing our paper. Thank you very much for your comment.

Best regards,
Kiara Ottogalli, Daniel Rosquete, Aiert Amundarain, Iker Aguinaga and Diego Borro

Reviewer 4 Report

Thank you very much for giving me an opportunity to review this manuscript. I did it with a great pleasure as the topic connects with my research interest. In my opinion the issue covered by the paper analysed is important and topical, especially in the age of Industry 4.0.

Authors  however, some additional efforts should made to improve the quality of the submission:

Title: Sugegestion of modification. Authors said in Abstract :

"This work proposes a new framework for modeling processes. It separates  the  process  definition  from  the  details  of  the  simulation  environment,  encompassing  the  most  important features from previous works, besides the flexibility of modeling industrial processes for different domains and purposes, interaction among human workers and automation systems, and independence  of  specific  programming  languages  or  technologies."

It seems that the intention of the authors was to propose a framework for modeling various organizational processes with the support of technologies typical of the Industry 4.0 era. (especially VR and AR)

More suitable title

Modeling of processes in the age of Industry 4.0 - flexible framework of virtual simulations

Abstract has a lack of:

-purpose of the study (The goal of this work is to propose a framework to model industrial processes in Virtual and  Augmented Reality simulators),

-the main methods or treatments applied (A critical literature review carried out using the desk research method and visualization of the proposed model based on characteristics of three case studies),

-and the article's main findings,  the main conclusions or interpretations.

Introduction and State of th art: a review of the literature in these parts should be in-depth, based on a broader review of the latest literature. It's worth looking for some new items

The research methodology is very poorly described. It is difficult to know how the overview of empirical  works  was carried out according  to  different  features and how the occurrence of the examined features was assessed in the research sample.(we see only results in Table1)

Assessment of authors' model (OWN in Table 1,line151) is too optimistic. In ihe Table 1 we can see that this model, as a one and only, is ideal for modelling o multiple processes. I am not sure it is tru. A lot of researchers give still innovative proposals. Reviewed only 34 literure items don't give a chance to be unambiguously sure and corectness of authors' assessment. Please,try to rethink and decrease or soften an assessment of authors' model (Maybe it is worth entering the appropriate comment under Table 1). The "Discussion" is very simple, underdeveloped. Authors don't compare their model with others similar, don't give an empirical examples of models of others researchers for analysed 3 industrial scenarios.  Enriching discussioin with specific findings,recommendations and proposals for improving it  would significantly increase the value of work. Please check for some editorial errors.

I congratulate the authors for this not easy work of trying to  bridge several key areas and concepts of Industry 4.0,  and wish them the most success.

 Don't hesitate to create next own ideas, strategies and  tools, helpful in developing of organization operating in the age of Industry 4.0

Best regards

Reviewer

Author Response

Dear reviewer,

We would like to thank you for your effort in reviewing our paper. Your comments have served to improve the paper. Below you will find the revisions we have made according to your comments.

Title: Sugegestion of modification. Authors said in Abstract : "This work proposes a new framework for modeling processes. It separates the  process  definition  from  the  details  of  the  simulation  environment,  encompassing  the  most  important features from previous works, besides the flexibility of modeling industrial processes for different domains and purposes, interaction among human workers and automation systems, and independence  of  specific  programming  languages  or  "

It seems that the intention of the authors was to propose a framework for modeling various organizational processes with the support of technologies typical of the Industry 4.0 era. (especially VR and AR)

More suitable title:

Modeling of processes in the age of Industry 4.0 - flexible framework of virtual simulations

Thank you very much for the suggestion related to the title, we did change it a little to make it more suitable to the purpose of our work based on your comment. We developed a framework to model industry 4.0 processes for VR/AR simulators, not to model the processes with these technologies, so we changed the title from “Flexible Framework for Modeling Industry 4.0 Processes in Virtual Simulators” to “Flexible Framework to Model Industry 4.0 Processes for Virtual Simulators”.

Abstract has a lack of:

-purpose of the study (The goal of this work is to propose a framework to model industrial processes in Virtual and  Augmented Reality simulators),

-the main methods or treatments applied (A critical literature review carried out using the desk research method and visualization of the proposed model based on characteristics of three case studies),

-and the article's main findings,  the main conclusions or interpretations.

Thanks to this comment we clarified the purpose of the work in the abstract, added the research method and emphasized on the findings.

Introduction and State of th art: a review of the literature in these parts should be in-depth, based on a broader review of the latest literature. It's worth looking for some new items

We added additional references of works presented in 2019.

The research methodology is very poorly described. It is difficult to know how the overview of empirical works  was carried out according  to  different  features and how the occurrence of the examined features was assessed in the research sample.(we see only results in Table1)

We explained better how the overview of the different works was made.

Assessment of authors' model (OWN in Table 1,line151) is too optimistic. In ihe Table 1 we can see that this model, as a one and only, is ideal for modelling o multiple processes. I am not sure it is tru. A lot of researchers give still innovative proposals. Reviewed only 34 literure items don't give a chance to be unambiguously sure and corectness of authors' assessment. Please,try to rethink and decrease or soften an assessment of authors' model (Maybe it is worth entering the appropriate comment under Table 1).

Effectively you are right on this point. We deleted the row and explained better the limitations of the framework.

The "Discussion" is very simple, underdeveloped. Authors don't compare their model with others similar, don't give an empirical examples of models of others researchers for analysed 3 industrial scenarios.

We improved the discussion section. Regarding the comparison, we understand your concern, but as what we are proposing is a framework, to make a comparison, we made a compendium of features that we presented in the summary table and associated explanation in Section 2. We grouped the features already present and new features (as considering many automation systems, and tasks besides assembly/disassembly) in our framework. Furthermore, to validate the generality of our framework, we present 3 different industrial applications of our framework used as a base for modelling simulations in both, VR and AR (Section 4).

Enriching discussioin with specific findings,recommendations and proposals for improving it would significantly increase the value of work. Please check for some editorial errors.

We improved the discussion section. We added recommendations and proposals. Also checked for errors.

I congratulate the authors for this not easy work of trying to bridge several key areas and concepts of Industry 4.0,  and wish them the most success. Don't hesitate to create next own ideas, strategies and  tools, helpful in developing of organization operating in the age of Industry 4.0

Thank you very much for your interest in our work and your comments, we found them very constructive.

Best regards,
Kiara Ottogalli, Daniel Rosquete, Aiert Amundarain, Iker Aguinaga and Diego Borro

Reviewer 5 Report

The article is an interesting study addressing the need & problems of designing new concepts of flexible framework for modeling in the era of Industry 4.0. The article discusses (major for the topic)  theoretical assumptions and then clearly illustrates them with practical examples (3 case studies). The entire argumentation in the article is conducted in an orderly and logical manner and easy to understand. Despite describing complicated issues, the authors managed to present it in a "reader-friendly" and interesting way. Also I would like to commend "2. State of the art” part of article - it proves knowledge of the subject and good substantive preparation of the authors, as well as the performance of satisfying desk research (34 publications, many of which were published after 2015).

If I were to point out some shortcomings of the article it would be a rather superficial / brief commentary on the summary presented in Table 1. The article would also gain if the authors devoted some attention to identifying weaknesses of the proposed solution, or challenges / requirements related to its implementation. A comment on encountered problems (during the design of the presented proposal) and the direction of further research on the topic would also be valuable.

Nevertheless, I think the article is very good and fully satisfying. I recommend for publication in an unchanged form, not counting the need to make a technical correction (e.g. remove breaks between individual paragraphs and removal of the yellow background of the text…).

Job well done ?

Author Response

Dear reviewer,

We would like to thank you for your effort in reviewing our paper. Your comments have served to improve the paper. Below you will find the revisions we have made according to your comments.

The article is an interesting study addressing the need & problems of designing new concepts of flexible framework for modeling in the era of Industry 4.0. The article discusses (major for the topic) theoretical assumptions and then clearly illustrates them with practical examples (3 case studies). The entire argumentation in the article is conducted in an orderly and logical manner and easy to understand. Despite describing complicated issues, the authors managed to present it in a "reader-friendly" and interesting way. Also I would like to commend "2. State of the art” part of article - it proves knowledge of the subject and good substantive preparation of the authors, as well as the performance of satisfying desk research (34 publications, many of which were published after 2015).

Thank you very much for your comments.

If I were to point out some shortcomings of the article it would be a rather superficial / brief commentary on the summary presented in Table 1.

Thanks to this comment we added the shortcomings of the framework in section 2.

The article would also gain if the authors devoted some attention to identifying weaknesses of the proposed solution, or challenges / requirements related to its implementation.

We added the weaknesses of the framework, and propose some challenges as future research in Section 5.

A comment on encountered problems (during the design of the presented proposal) and the direction of further research on the topic would also be valuable.

Added a relevant problem we encountered in section 5. Also added comments about future search directions on section 5.

Nevertheless, I think the article is very good and fully satisfying. I recommend for publication in an unchanged form, not counting the need to make a technical correction (e.g. remove breaks between individual paragraphs and removal of the yellow background of the text…).

Job well done ?

Thank you again for all your comments, we are glad you found it satisfying.

Best regards,
Kiara Ottogalli, Daniel Rosquete, Aiert Amundarain, Iker Aguinaga and Diego Borro

This manuscript is a resubmission of an earlier submission. The following is a list of the peer review reports and author responses from that submission.

Round 1

Reviewer 1 Report

The topic could be interesting and timely. However the paper has several weaknesses. E.g.

The term “Industry 4.0” seems to appear in the article just because it is fashionable. In fact, the arguments presented in the first sentence of the abstract are trivial and long existing in industry …. Not only in Industry 4.0. Therefore, if the authors want to focus on Industry 4.0 then a deeper argumentation is needed. Line 71 does not start in a good way. The Introduction should be revised namely because some features presented as new were already present, to some extent, in previous simulators. Thus, some more accurate description is needed (to not give the impression of re-inventing the wheel). The authors have a number of new contributions and should focus on them without ignoring what was done before. For instance, the example of simulation of assembly tasks has been extensively addressed before. It would also be better to separate the Introduction from the Related work. In current version everything is merged in the same section. Along the paper (PDF version) there are MANY errors like “¡Error! No se encuentra  el  origen  de  la  ”. This makes it difficult to follow the paper. The authors should have paid attention to this. In section 2 the authors need to characterize / identify from the beginning which types of processes they have in mind. Please be aware that “processes” are different in discrete manufacturing and process industries. In Fig. 2 the attributes of process seem quite limited (just an id). What if a task does not have sub-tasks? Do you mean that a task must always have at least one sub-task? Clarify the difference between actor and agent. Section 2, surprisingly, does not mention anything about AR/VR (that was “promised” in the Introduction). In section 3 the authors show some proof-of-concept – ONLY in terms of showing that “it can be done”, but there is no validation! How does this solution compare to others? What / where are the innovative features? How well they perform? Also the Discussion section is rather limited regarding validation. Since simulation is an “old” topic, the reader would expect that the authors focused on the novel features and clearly show them (through validation). Otherwise, many of the initial claims are not proved at all.

Therefore, I can only suggest that the authors deeply revise the paper, prepare a serious validation part and resubmit it as a new submission.

Reviewer 2 Report

Authors of this article are trying to formulate 'framework' for handling complicated situations/problems that arise as a result of 'Industry 4.0 revolution'. I greatly appreciate the intention and effort of authors, but mentioned 'attitude' is already known to every programmer. I believe that there is necessary to create/formulate some kind of 'model theory' a based on that there can be this type of framework formulated. 

I realize that creating of this type of framework is not easy task, but I not sure that this one, which is formulated in this paper is sufficient (even when we consider different states and methods of existing one)..

Reviewer 3 Report

Paper is good organized. Flexible lines are at very actual.

Authors has represent all what is required for scientific paper, starting from Abstract, Introduction, Methodology and results. Conclusions are derived based on results. 

Reviewer 4 Report

There are some problems at lines 71 (starting with a numbered reference), line 135 (error message), line 139 (error message), line 195 (error message), line 237 (error message), line 384 (error message) and line 411 (error message)

Line 414 real time ergonomics assessment is mentionned but no details are provided. What about safety associated with the robot near the worker (being injured by the robot)? Does the simulator accounts for robotic safety?

Industry 4.0 is in the title but not really in the article. Including additional references on virtual reality and augmented reality as design tools in an industry 4 era might help.

The reference list can be made stronger